# Effects of α₂-adrenoceptor stimulation on luminal alkalinisation and net fluid flux in rat duodenum

Olof Nylander◉, Markus Sjöblom◉, John Sedin, David Dahlgren◉*

Division of Physiology, Department of Medical Cell Biology, Uppsala University, Uppsala, Sweden

◉ These authors contributed equally to this work.
* david.dahlgren@farmbio.uu.se

**Data Availability Statement:** All relevant data are within the paper and its Supporting Information files.

**Funding:** The author(s) received no specific funding for this work.

## Abstract

The sympathetic nervous system is highly involved in the regulation of gastrointestinal functions such as luminal alkalinisation and fluid absorption. However, the exact mechanisms are not clear. This study aimed to delineate how α₂-adrenergic receptor stimulation reduces duodenal luminal alkalinisation and induces net fluid absorption. This was tested by perfusing the duodenum of anesthetized rats with isotonic solutions devoid of $Cl^-$ and/or $Na^+$, in the absence and presence of the α₂-adrenoceptor agonist clonidine. The clonidine was also studied in rats treated with dimethylamiloride (a $Na^+/H^+$ exchange inhibitor), vasoactive intestinal peptide, and the nicotinic receptor antagonist hexamethonium. Clonidine reduced luminal alkalinisation and induced net fluid absorption. The $Cl^-$-free solution decreased luminal alkalinisation and abolished net fluid absorption, but did not prevent clonidine from doing so. Both the $Na^+$-free solution and luminal dimethylamiloride increased luminal alkalinisation and abolished net fluid absorption, effects counteracted by clonidine. The NaCl-free solution (D-mannitol) did not affect luminal alkalinisation, but reduced net fluid absorption. Clonidine reduced luminal alkalinisation and induced net fluid absorption in rats perfused luminally with mannitol. However, clonidine did not affect the vasoactive intestinal peptide-induced increase in luminal alkalinisation or fluid secretion. Pre-treatment with hexamethonium abolished the effects of clonidine on luminal alkalinisation and net fluid flux. In summary, our *in vivo* experiments showed that clonidine-induced reduction in luminal alkalinisation and induction of net fluid absorption was unrelated to luminal $Na^+$ and $Cl^-$, or to apical $Na^+/H^+$ or $Cl^-/HCO_3^-$ exchangers. Instead, clonidine seems to exert its effects via suppression of nicotinic receptor-activated acetylcholine secretomotor neurons.

## Introduction

Sympathetic postganglionic nerve fibers enter the intestinal wall along arteries. These fibers terminate primarily in the myenteric and submucosal plexuses, but some also penetrate into the submucosa and mucosa [1]. Upon activation of these neurons, noradrenalin is released, which binds to two main types of adrenergic receptors, α and β. This activation inhibits peristalsis, reduces blood flow, and increases fluid absorption.

**Competing interests:** The authors have declared that no competing interests exist.

Adrenergic enteric neurons induces net fluid absorption by stimulation of absorption and/or inhibition of fluid secretion via an $\alpha_2$-adrenoceptor mediated mechanism [2]. This may be attributed to $\alpha_2$-adrenoceptor-induced inhibition of cholinergic nerves in the myenteric plexus or via a direct action on epithelial cells, or both [3]. The antisecretory effect may be direct, or it may be indirect as a result of reduced gut motility [4]. The pro-absorptive effect may be related to activation of apical ion transporters in the villus epithelium or inhibition of cystic fibrosis transmembrane regulator (CFTR, Abc35) or other $Cl^-$-channels in the crypt epithelium. However, despite extensive studies [5–13], the exact mechanism by which $\alpha_2$-adrenoceptor stimulation affects electrolyte-fluid flux in the duodenum remains unclear.

The duodenal mucosa of several species, including humans, transports bicarbonate ($HCO_3^-$) into the luminal solution at a considerable rate. This is achieved via CFTR and apical $Cl^-/HCO_3^-$ exchangers: downregulated in adenoma (DRA, Slc26a3), putative anion transporter 1 (PAT-1, Slc26a6), and anion exchanger isoform 4 (AE4, Slc4a9). Bicarbonate also enters the lumen, to a much lesser extent, by passive diffusion through paracellular pathways [14, 15]. To a small extent, secretion of $H^+$ by the apical $Na^+/H^+$ exchangers, mainly NHE3 (Slc9a3) reduces luminal alkalinisation in human [16], rat [17], and mouse duodenum [18]. The rate of duodenal luminal alkalinisation is regulated by the autonomic nervous system, including the enteric nervous system, paracrine factors, and hormones [19]. Previous *in vivo* experiments have shown that duodenal mucosal alkaline secretion is reduced by electrical stimulation of the sympathetic splanchnic nerves in rat [20] and cat [21], and following intravenous injection of clonidine, a potent $\alpha_2$-adrenoceptor agonist, in rats [22, 23], and humans [24]. Currently, the interplay between the above-mentioned transporters and $\alpha_2$-adrenoceptor inhibition has not been investigated *in vivo*.

The aim of the present investigation was to further delineate the mechanism by which $\alpha_2$-adrenergic receptor stimulation by clonidine reduces duodenal luminal alkalinisation and induces net fluid absorption in rats *in vivo*. More specifically, we wanted to answer if (i) the effects of clonidine on electrolyte and water transport were due to increased absorption or reduced secretion or a combination of both, (ii) if the effects of clonidine were sensitive to the removal of luminal $Cl^-$ and/or $Na^+$, and (iii) if clonidine exerted its effects via suppression of excitatory nicotinergic receptor-activated secretomotor neurons.

The influence of ion transporter activity in the epithelial brush border membrane (i.e., the $Cl^-/HCO_3^-$ and $Na^+/H^+$ exchangers, and CFTR) was evaluated by perfusing the duodenum with $Cl^-$ or $Na^+$ free solutions, and with or without clonidine. We also investigated the effects of clonidine in animals pre-treated with the NHE-inhibitor dimethylamiloride (DMA), and the non-selective, nicotinic-acetylcholine receptor antagonist hexamethonium. The latter drug has been shown to reduce basal duodenal luminal alkalinisation and to abolish the increase in $HCO_3^-$ secretion elicited by electrical stimulation of the vagal nerve, suggesting inhibition of enteric excitatory neurons [25, 26]. Finally, we evaluated whether clonidine affected the vaso-active intestinal peptide (VIP) induced stimulation of electrolyte fluid secretion. VIP-induced increase in luminal alkalinisation and fluid secretion is absent in CFTR-knockout mice, suggesting that VIP exerts its effect on secretion via activation of CFTR [27].

## Materials and methods

### Animals and surgery

The material in this study is in conformity with Good Publishing Practice in Physiology [28]. The study was approved by the local ethics committee for animal research (no: C250/12) in Uppsala, Sweden. Male Sprague Dawley rats (n = 105) weighing from 260–389 g (mean ± SD: 316 ± 27 g) were purchased from Scanbur AB, Sollentuna, Sweden. The animals were

maintained under constant conditions (12:12 h light-dark cycles; 21˚C) with ad libitum access to food and water. Before the experiments, the rats were fasted (in pairs) overnight with free access to water. Thereafter they were anaesthetized with an intraperitoneal injection of 125 mg kg$^{-1}$ thiobutabarbital sodium salt (Inactin, St. Louis, MO, USA). Body temperature was maintained at 37.5 ± 0.5˚C during the surgical procedure. The single-pass duodenal perfusion experiment was the same as described in [29]. At the end of the perfusion experiment, rats were sacrificed with a i.v. injection with saturated KCl solution.

## Measurement of duodenal luminal alkalinisation

The luminal alkalinisation was assessed by back titration as described previously [30], and expressed as micromoles of base transported per cm$^2$ serosal surface area per hour (µmol cm$^{-2}$ h$^{-1}$).

## Measurement of fluid flux

The method to assess transepithelial net fluid flux is detailed in [30]. In brief, the absolute flux was determined by subtracting the collected effluent volume from the peristaltic pump volume. The net change in fluid flux in response to the test solution was determined as follows. The mean of the effluent volumes sampled before the exposure to the test solution was subtracted from the mean value in response to the test solution, in relationship to the weight of the duodenum as determined after the experiment. Fluid flux was expressed in ml per g wet tissue weight per hour (ml g$^{-1}$ h$^{-1}$). The drift of the peristaltic pump over time was insignificant (<0.1%).

## Experimental protocol

The single-pass intestinal perfusion setups with treatments and luminal conditions are shown in Fig 1. Mean arterial blood pressure (MABP), the rate of luminal alkalinisation and the transepithelial net fluid flux were all assessed.

## Clonidine with and without idazoxan

The duodenum was perfused luminally with a 155 mM NaCl solution throughout the 90-min experiment. Thirty min after the start of the experiment, clonidine was administered

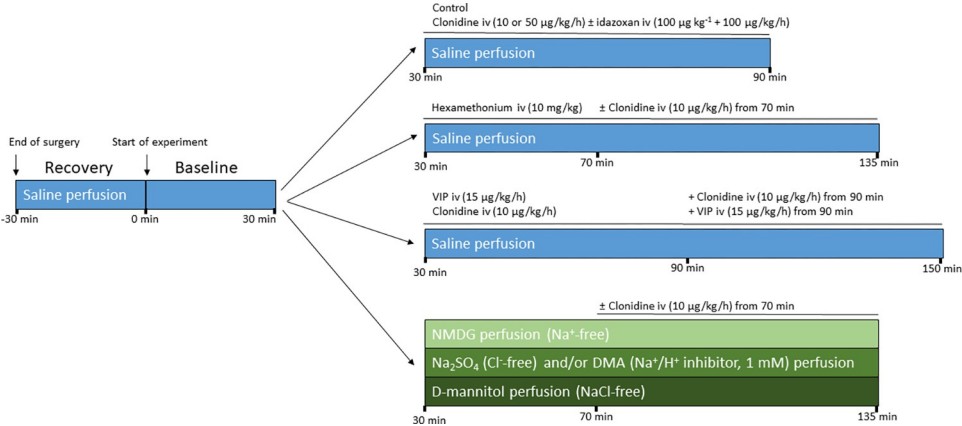

**Fig 1. Experimental setups of the intestinal perfusion.** In all rat groups, the duodenum was initially perfused (0.4 mL/min) with isotonic saline (blue) for 30 min (stabilization period) followed by a 30 min period to assess basal values of luminal alkalinisation and net fluid flux. Each group was thereafter perfused with isotonic saline for 60, 105 or 120 min, or with a solution free from Na$^+$ (light green), Cl$^-$ (green), or NaCl (dark green). Each of these groups were tested alone, and after intravenous clonidine treatment. In one set of experiments, clonidine was also tested with or without intravenous VIP, and in another with or without luminal DMA, a Na$^+$/H$^+$ exchange inhibitor.

intravenously as a continuous infusion (1.0 ml h$^{-1}$) at a dose of 10 or 50 µg kg$^{-1}$ h$^{-1}$. A third subgroup used the same protocol as the clonidine 10 µg kg$^{-1}$ h$^{-1}$ except that idazoxan, an $\alpha_2$-adrenoceptor antagonist, was administered intravenously (100 µg kg$^{-1}$ bolus + 100 µg kg$^{-1}$ h$^{-1}$ infusion) starting 30-min before the start of the experiment.

### Perfusion with a Cl$^-$-free Na$_2$SO$_4$ solution

After the initial 30-minute isotonic NaCl perfusion, the duodenum was perfused with a Cl$^-$-free isotonic Na$_2$SO$_4$ solution (75 mM Na$_2$SO$_4$ with 60 mM D-mannitol) for 100 min. The protocol was the same for the second subgroup except that clonidine was given by an intravenous infusion (1 ml h$^{-1}$) at a dose of 10 µg kg$^{-1}$ h$^{-1}$, starting 40 min after commencement of the Na$_2$SO$_4$ perfusion and continued for 60 min.

### Perfusion with a Na$^+$-free NMDG solution

The experimental protocol was exactly the same as for the Cl$^-$-free perfusion (Na$_2$SO$_4$), except that the solution was a *Na$^+$-free* isotonic N-methyl-D-glucamine chloride (NMDG-Cl; 155 mM; 285–291 mOsm kg$^{-1}$ H$_2$O).

### Perfusion with a Na$^+$/H$^+$ exchange inhibitor, with and without luminal Cl$^-$

After the 30-min perfusion with isotonic NaCl, the duodenum was perfused for 100 min with the Na$^+$/H$^+$ exchange inhibitor DMA (1 mM) in either an isotonic NaCl solution or the Cl$^-$-free isotonic solution described above. The protocol for the third subgroup was the same except that clonidine was administered intravenously starting 40 min after commencement of the perfusion with the DMA or the Cl$^-$-free DMA solution and continued for 60 min.

### Perfusion with a NaCl-free D-mannitol solution

After an initial 30-min period of isotonic NaCl perfusion, the duodenum was perfused with an isotonic D-mannitol solution (260 mM D-mannitol; 286–290 mOsm kg$^{-1}$ H$_2$O). In the second subgroup, clonidine was administered intravenously at a dose of 10 µg kg$^{-1}$ h$^{-1}$ starting 40 min after commencement of the mannitol perfusion and continued for 60 min.

### Effects of vasoactive intestinal peptide

Animals were divided in two subgroups. In both groups, the duodenum was perfused with isotonic NaCl throughout the experiment. In the first subgroup, vasoactive intestinal peptide (VIP) was intravenously infused at a rate of 15 µg kg$^{-1}$ h$^{-1}$ starting 30 min after commencement of effluent collection and continued for 120 min. Sixty min after the start of the VIP infusion, clonidine was continuously infused at 10 µg kg$^{-1}$ h$^{-1}$ throughout the experiment. In the second subgroup, the clonidine was administered 30 min after start of effluent collection. Sixty min after commencement of the clonidine infusion, VIP was administered intravenously at 15 µg kg$^{-1}$ h$^{-1}$ throughout the experiment.

### Effects of hexamethonium

The duodenum was perfused with isotonic NaCl throughout the experiment. Thirty min after initiation of the perfusion, hexamethonium, a non-selective competitive nicotinic receptor antagonist, was administered intravenously as a bolus at 10 mg kg$^{-1}$ followed by a continuous infusion at 10 mg kg$^{-1}$ h$^{-1}$. In the second subgroup, clonidine was given by an intravenous infusion (1 ml h$^{-1}$) at a dose of 10 µg kg$^{-1}$ h$^{-1}$, starting 40 min after start of the hexamethonium infusion and continued for 60 min.

## Chemicals

Bovine albumin, DMA, D-mannitol, idazoxan hydrochloride, Inactin, hexamethonium chloride and VIP were purchased from Sigma-Aldrich (St. Louis, MO, USA). Clonidine HCl was purchased from Tocris Bioscience (Bristol, UK). NaCl, $Na_2SO_4$, and NMDG were purchased from Merck, Darmstadt, Germany.

## Statistical analyses

Values are expressed as means ± SEM. The statistical significance of the data was tested by analysis of variance (ANOVA) followed by Tukey's Multiple Comparison test. To test differences within a group, i.e. comparing the results obtained before, during, and after perfusion with the different solutions, a one-factor repeated measures ANOVA was used. Differences between two groups of animals was tested by students *t*-test, and when multiple comparisons were needed an unpaired two-factor repeated measures ANOVA was used. All statistical analyses were performed using GraphPad Prism software. P<0.05 was considered as significant (two-tailed test). The data that support the findings of this study are available from the corresponding author upon reasonable request.

# Results

## Basal luminal alkalinisation and fluid flux

The duodenum in all groups was perfused with isotonic NaCl for 30 min. The mean basal rate of luminal alkalinisation was 7.1 ± 2.7 µmol cm$^{-2}$ h$^{-1}$ and the basal net fluid flux was -0.81 ± 1.18 ml g$^{-1}$ h$^{-1}$ (mean ± SD, n = 105 for both). The net fluid flux was significantly below zero (i.e. net fluid absorption, P<0.001), which is depicted with a minus sign in text and figures. There was no linear correlation ($r^2$ = 0.02, P = 0.13) between the basal absolute rate of luminal alkalinisation and the basal net fluid flux.

## Effect of $\alpha_2$-adrenoceptor stimulation on basal parameters

The effects of clonidine, a well-known α2-adrenoceptor agonist, was studied on duodenal fluid flux, duodenal luminal alkalinisation and mean arterial blood pressure. Intravenous infusion of clonidine at a dose of 10 µg kg$^{-1}$ h$^{-1}$ significantly (P<0.001) reduced the MABP and duodenal luminal alkalinisation (P<0.001), and induced net fluid absorption (P<0.001) (Fig 1A–1C). Clonidine at 50 µg kg$^{-1}$ h$^{-1}$ induced virtually the same results on luminal alkalinisation and net fluid flux as did 10 µg kg$^{-1}$ h$^{-1}$ (Fig 1D and 1E), but the decrease in MABP was faster and less pronounced (P<0.05) (Fig 2F). The magnitude of the clonidine-induced decrease in luminal alkalinisation and the change in net fluid flux were both linearly correlated (P<0.001) to basal luminal alkalinisation (y = 1.08–0.61x, $r^2$ = 0.74, n = 21) and basal net fluid flux (y = -1.49–0.52x, $r^2$ = 0.64, n = 21), respectively (Fig 1G and 1H).

## Effect of $\alpha_2$-adrenoceptor inhibition on basal parameters

To examine whether clonidine affected basal parameters by stimulation of α2-adrenoceptors, clonidine was tested in animals pretreated with the α2-adrenoceptor antagonist idazoxan. The decrease was significantly lower (P<0.05) in idazoxan-treated animals (-1.2 ± 0.5 µmol cm-2 h-1, n = 5) than in controls (-3.0 ± 0.4 µmol cm-2 h-1, n = 13). Idazoxan abolished the pro-absorptive action of clonidine on net fluid flux (the net change was 0.13 ± 0.51 as compared to -1.48 ± 0.37 ml g$^{-1}$ h$^{-1}$ in animals treated with clonidine alone, P<0.05). The clonidine-induced decrease in MABP was significantly (P<0.001) lower in idazoxan-treated animals (-10 ± 2 mm Hg) than in the controls.

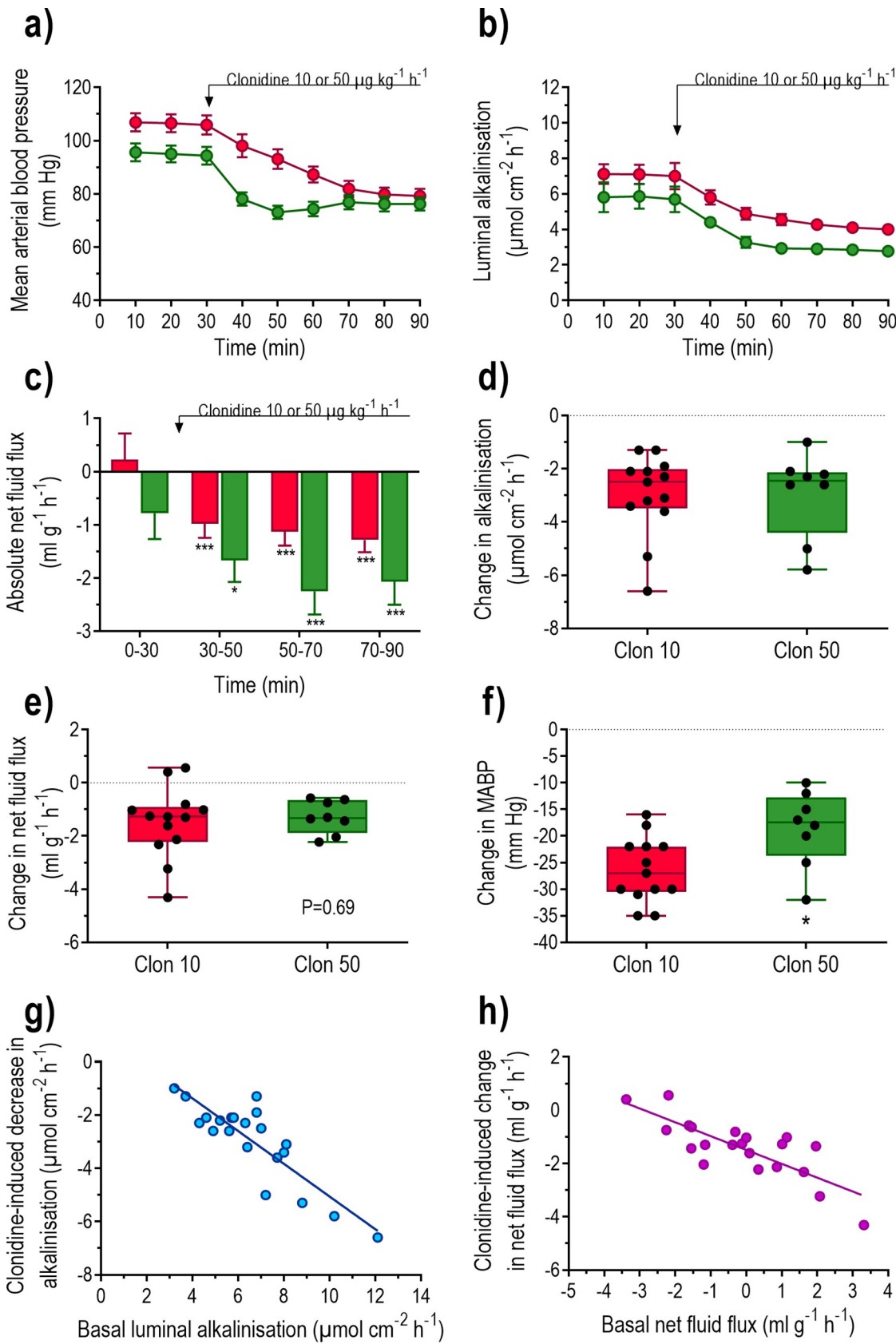

**Fig 2. The effects of clonidine on mean arterial blood pressure, fluid flux and luminal alkalinisation.** Duodenum was perfused with isotonic saline for 90 min and clonidine was administered from 30 min as a constant i.v. infusion at a dose of 10 or 50 $\mu$g kg$^{-1}$ h$^{-1}$ (Fig 1). Shown are the mean (a) arterial blood pressure (MABP), (b) rate of luminal alkalinisation, (c) transepithelial net fluid flux, (d) net change in luminal alkalinisation in response to clonidine, (e) net change in net fluid flux and (f) net change in MABP in response to clonidine. Relationship between the (g) basal luminal alkalinisation and the clonidine-induced decrease in alkalinisation (y = -0.65x + 1.60, $r^2$ = 0.77, P<0.001), and (h) basal net fluid flux and the clonidine-induced change in net fluid flux (y = -0.70x - 1.26, $r^2$ = 0.85, P<0.001). Values are means ± SEM or box plots with all individual points, n = 12. Changes are presented as the mean of values at 80 and 90 min minus the mean of the three control values. ***P<0.001 compared with basal values.

## Effect of α2-adrenoceptor stimulation in the absence of luminal Cl⁻

The aim was to determine whether the effects of clonidine on luminal alkalinisation and net fluid flux are dependent on apical chloride $HCO_3^-$ exchange. To examine this, the duodenum was perfused with an isotonic Cl⁻-free $Na_2SO_4$ solution in the absence and presence of clonidine. The isotonic Cl⁻-free $Na_2SO_4$ solution decreased (P<0.01) luminal alkalinisation and changed (P<0.001) net fluid flux from net absorption to values not different from zero (Fig 3A and 3B). Interestingly, the magnitude of the decrease in luminal alkalinisation in response to the Cl⁻-free $Na_2SO_4$ solution varied greatly between the animals and was linearly correlated to basal luminal alkalinisation (y = 1.68–0.64x, $r^2$ = 0.70, P<0.001 and n = 14) (Fig 4A).

In animals perfused with the Cl⁻-free $Na_2SO_4$ solution clonidine decreased (P<0.01) luminal alkalinisation further (by 52%) and induced (P<0.001) net fluid absorption (Fig 3C and 3D). The changes in luminal alkalinisation and net fluid flux were both significantly greater in the animals treated with $Na_2SO_4$ and clonidine than with the $Na_2SO_4$ alone (Fig 3E and 3F).

## Effect of α₂-adrenoceptor stimulation in the absence of luminal Na⁺

The objective was to determine whether the effects of clonidine on duodenal luminal alkalinisation and net fluid flux are dependent on apical Na⁺-H⁺ exchange. This was done by perfusion of the duodenum with an isotonic Na⁺-free NMDG chloride solution in the absence and presence of clonidine. This solution increased (P<0.05) luminal alkalinisation and changed (P<0.05) basal net fluid flux from net absorption to zero (Fig 5A and 5B). No linear correlation was found between the basal luminal alkalinisation and the NMDG-induced increase in luminal alkalinisation ($r^2$ = 0.19, P = 0.22, n = 10) (Fig 4B).

In animals perfused with an isotonic Na⁺-free NMDG chloride solution clonidine decreased (P<0.001) luminal alkalinisation and changed (P<0.01) net fluid flux from a value not different from zero to net fluid absorption (Fig 5C and 5D). The changes in luminal alkalinisation and the net fluid flux were both significantly greater in NMDG plus clonidine treated animals than in those treated with NMDG alone (Fig 5E and 5F).

## Effect of α₂-adrenoceptor stimulation in the presence of luminal dimethylamiloride

The aim was to determine whether the effects of clonidine on luminal alkalinisation and net fluid flux are affected by luminal dimethylamiloride (DMA), a non-specific inhibitor of Na⁺/H⁺ exchange. DMA increased luminal alkalinisation (P<0.05), and changed (P<0.01) net fluid flux from a basal value not different from zero towards net fluid secretion (Fig 6A and 6B). No linear correlation was found between the basal luminal alkalinisation and the DMA-induced increase in luminal alkalinisation ($r^2$ = 0.12, P = 0.25, n = 13) (Fig 4C).

In animals perfused with DMA clonidine significantly decreased the rate of luminal alkalinisation and induced net fluid absorption (Fig 6C and 6D). The changes in luminal alkalinisation and net fluid flux were both significantly greater in DMA plus clonidine treated rats than in those treated with DMA alone (Fig 6E and 6F).

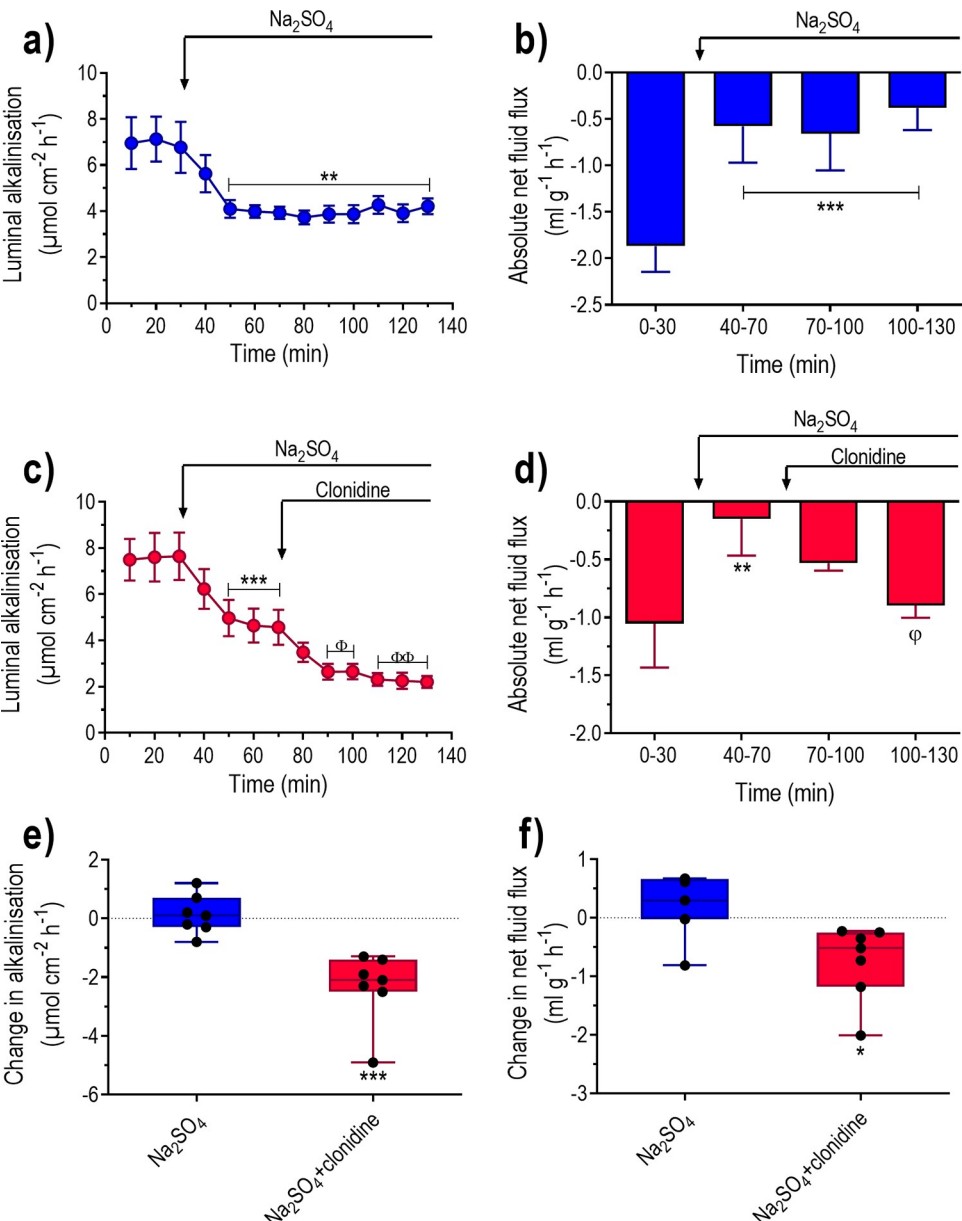

**Fig 3. The effects clonidine on net fluid flux and luminal alkalinisation in rat duodenum perfused with an isotonic Cl⁻-free solution.** Duodenum was perfused with isotonic saline for 30 min and subsequently with an isotonic Cl⁻-free $Na_2SO_4$ solution for 100 min (Fig 1). Effects on luminal alkalinisation (a and c) and transepithelial net fluid flux (b and d) in the absence and presence of i.v. infusion of clonidine at a dose of 10 μg kg⁻¹ h⁻¹. Net changes in (e) luminal alkalinisation and (f) transepithelial net fluid flux between 110–130 min and 50–70 min ($Na_2SO_4$ plus clonidine vs. $Na_2SO_4$ alone). Values are means ± SEM or box plots with all individual points. *P<0.05, **P<0.01 and ***P<0.001 compared with basal values. ᶲP<0.05 and ᶲᶲP<0.01 compared with values at 40–70 min. Fig (e) and (f); *P<0.05 and ***P<0.001 compared with values in animals treated with $Na_2SO_4$ alone.

## The effect of α2-adrenoceptor stimulation in the absence of luminal NaCl

The objective was to determine whether the effects of clonidine on luminal alkalinisation and net fluid flux is affected by the lack of luminal NaCl. To achieve this, the duodenum was perfused with an isotonic NaCl-free solution, i.e., an isotonic D-mannitol solution. The isotonic D-mannitol solution had no significant effect on the mean luminal alkalinisation (Fig 7a).

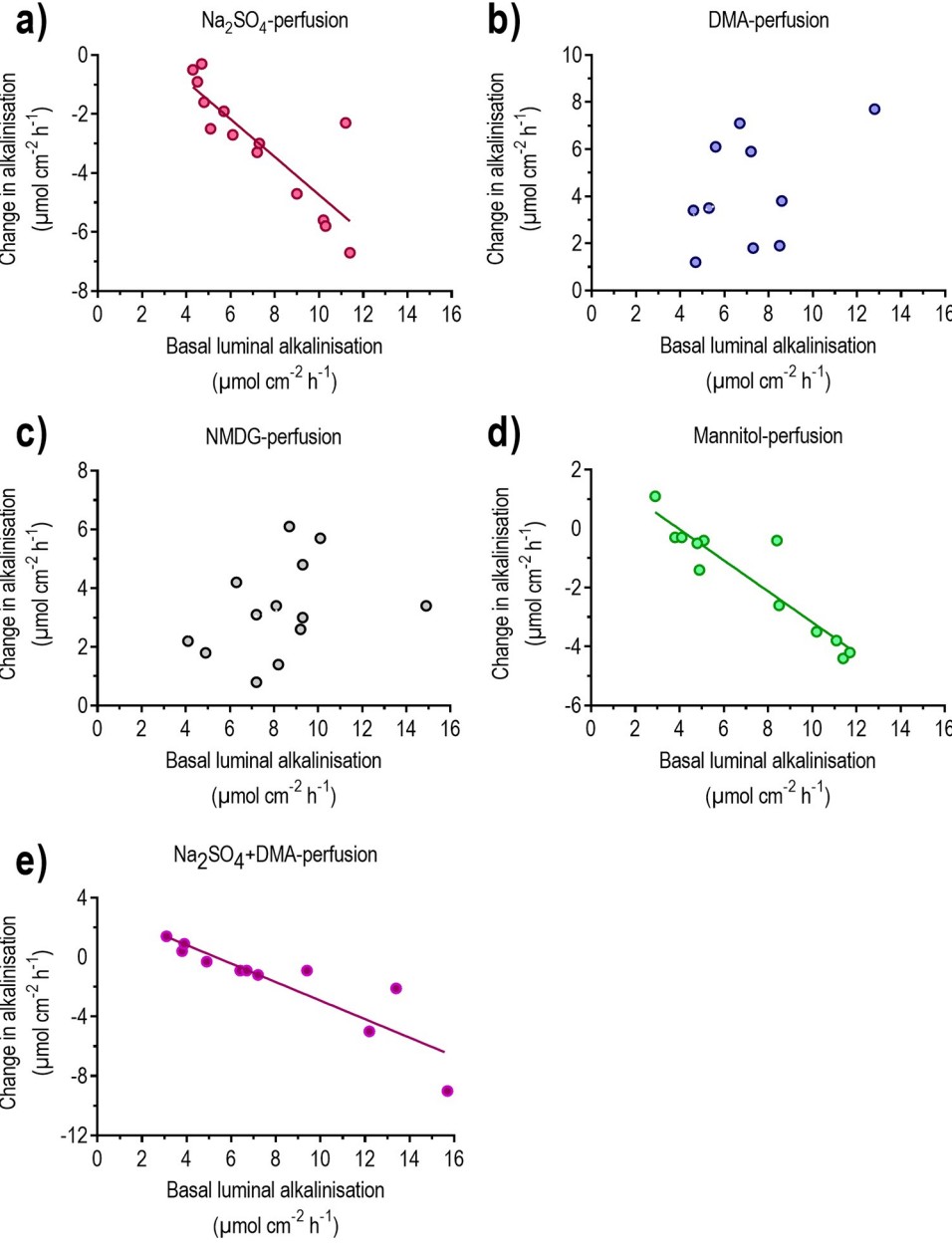

**Fig 4. The relationship between the changes in luminal alkalinisation in response to different luminal solutions and basal luminal alkalinisation.** The duodenum was perfused for 40 min with (a) Cl⁻-free $Na_2SO_4$, (b) Na⁺-free NMDG, (c) dimethylamiloride (DMA), (d) NaCl-free D-mannitol, or (e) Cl⁻-free $Na_2SO_4$ plus DMA. Each x-y value is the mean of three basal values before treatment and the mean of the two last values in response to treatment. Regression analysis: (a). y = -0.64x + 1.68, $r^2$ = 0.70, P<0.001, n = 14. (b) y = 0.21x + 1.55, $r^2$ = 0.12, P = 0.25, n = 13. (c). y = 0.40x + 1.32, $r^2$ = 0.19, P = 0.21, n = 10. (d). y = -0.52x + 2.08, $r^2$ = 0.86, P<0.001, n = 12. (e). y = -0.62x + 3.31, $r^2$ = 0.79, P = 0.001, n = 11.

However, a very good (P<0.001) linear correlation (y = 2.08–0.52x, $r^2$ = 0.86) was found between the basal luminal alkalinisation and the mannitol-induced change in luminal alkalinisation (Fig 4D). During the perfusion with isotonic NaCl there was a net fluid absorption, which decreased significantly in response to the isotonic mannitol solution (Fig 7B). The

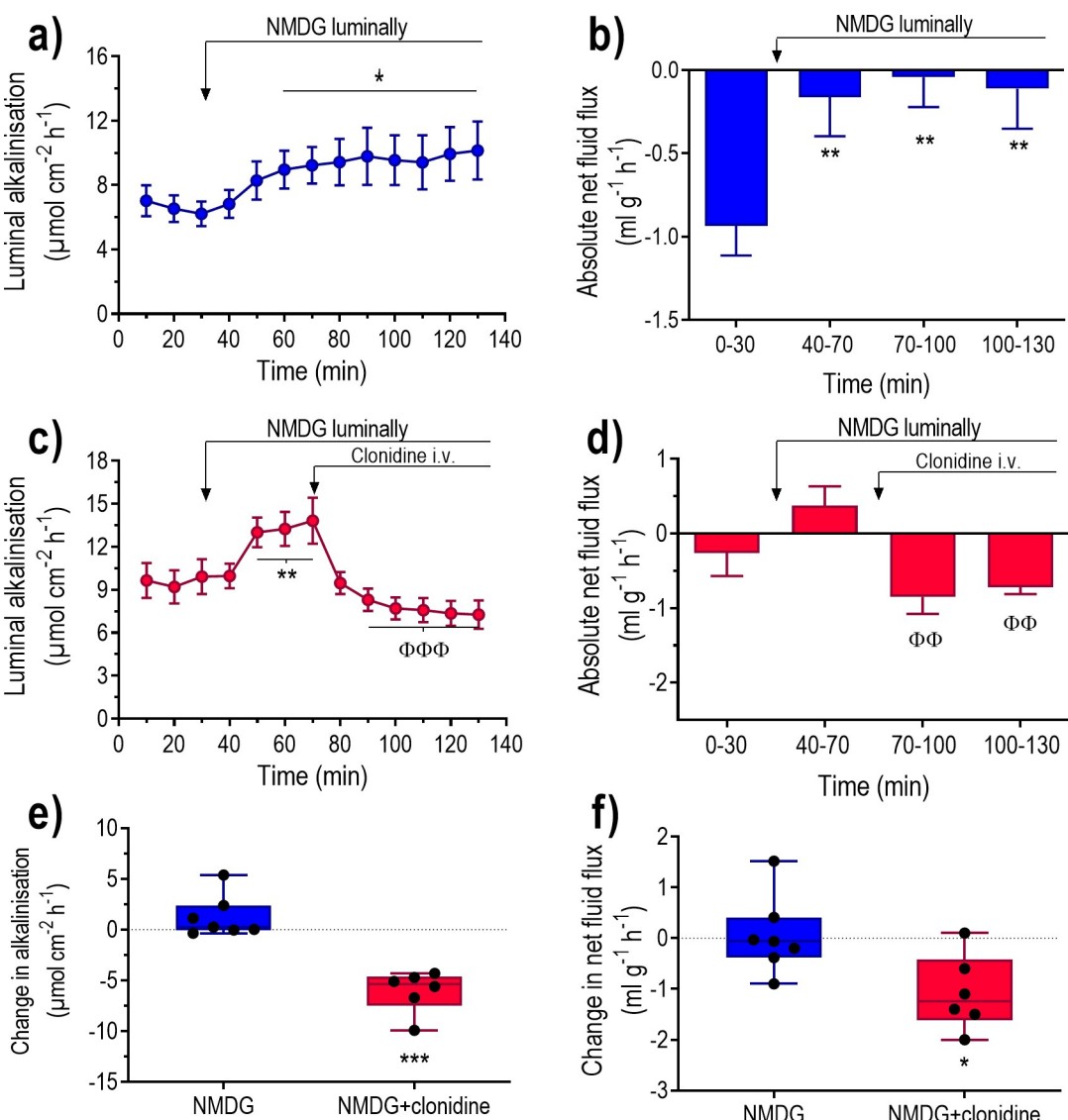

**Fig 5. The effects of clonidine on fluid flux and luminal alkalinisation in rat duodenum perfused with an isotonic Na⁺-free solution.** The duodenum was perfused with isotonic saline for 30 min, followed by an isotonic Na⁺-free NMDG chloride solution for 100 min (Fig 1). Effects on luminal alkalinisation (a and c) and transepithelial net fluid flux (b and d) in the absence and presence of i.v. infusion of clonidine at a dose of 10 μg kg⁻¹ h⁻¹. Net changes in (e) luminal alkalinisation and (f) transepithelial net fluid flux between 110–130 min and 50–70 min (NMDG plus clonidine vs. NMDG alone). Values are means ± SEM or box plots with all individual points. *P<0.05 and **P<0.01 compared with basal values. ᵠᵠP<0.01 and ᵠᵠᵠP<0.001 compared with values at 40–70 min. Fig (e) and (f); *P<0.05 and ***P<0.001 compared with values in animals treated with NMDG alone.

absolute net fluid flux at 100–130 min was significantly below zero (P<0.02), i.e., net fluid absorption.

In animals perfused with the isotonic NaCl-free solution clonidine significantly decreased luminal alkalinisation and increased net fluid absorption (Fig 7C and 7D). The changes in luminal alkalinisation and the net fluid flux were both significantly greater in mannitol plus clonidine treated animals than in mannitol treated ones (Fig 7E and 7F).

**The effects of α₂-adrenoceptor stimulation in the absence of luminal Cl⁻ and in the presence of dimethylamiloride.** To further examine whether the effects induced by clonidine

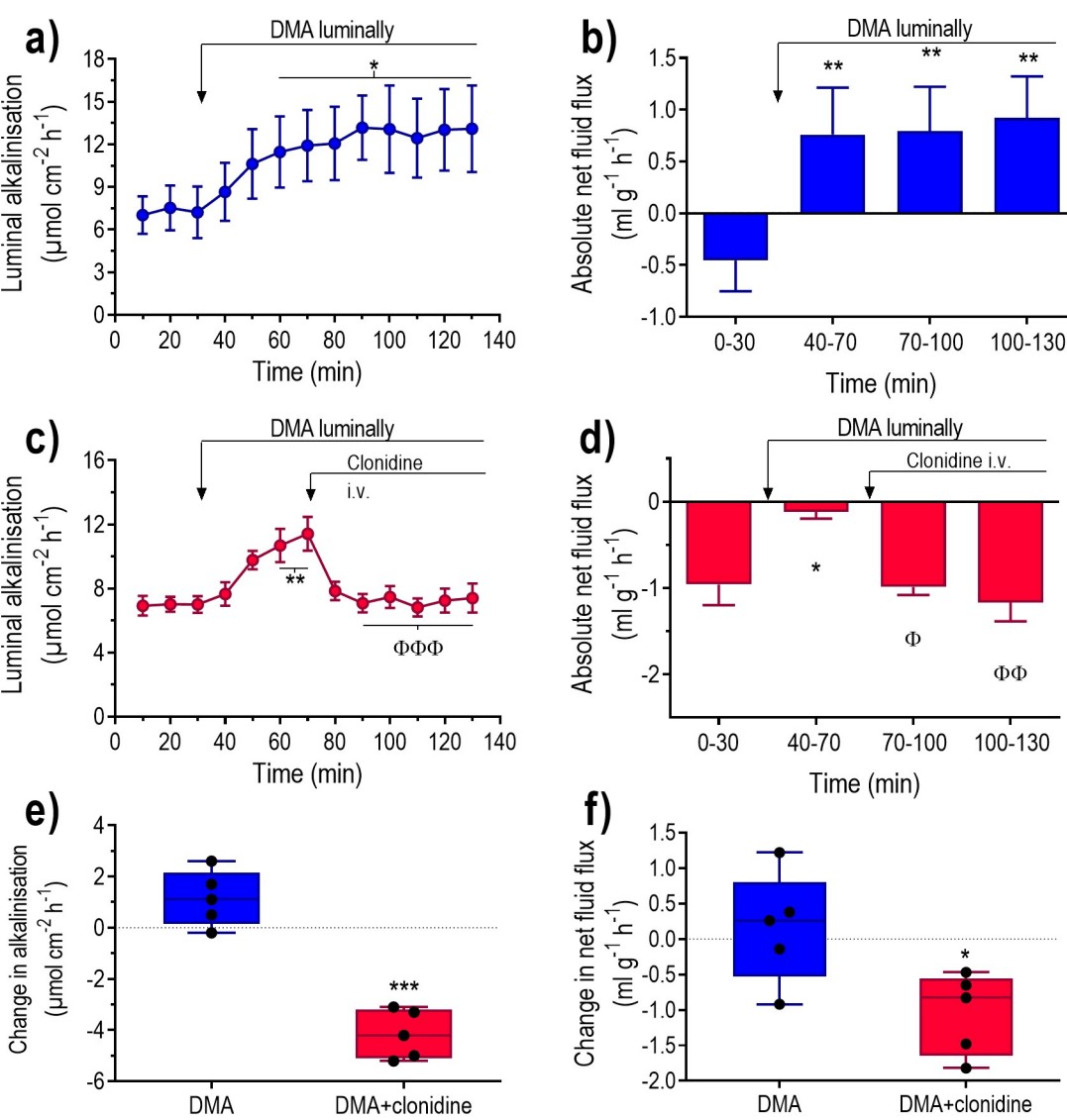

**Fig 6. The effects of clonidine on fluid flux and luminal alkalinisation in the rat duodenum perfused with DMA.**
Duodenum was perfused with isotonic saline for 30 min and subsequently with isotonic dimethylamiloride (DMA) solution ($10^{-3}$ M) for 100 min (Fig 1). Effects on luminal alkalinisation (a and c) and transepithelial net fluid flux (b and d) in the absence and presence of i.v. infusion of clonidine at a dose of 10 µg kg$^{-1}$ h$^{-1}$. Net changes in (e) luminal alkalinisation and (f) transepithelial net fluid flux between 110–130 min and 50–70 min (DMA plus clonidine vs DMA alone). Values are means ± SEM or box plots with all individual points. $^*$P<0.05 and $^{**}$P<0.01 compared with basal values. $^{\Phi}$P<0.05, $^{\Phi\Phi}$P<0.01 and $^{\Phi\Phi\Phi}$P<0.001 compared with values at 50–70 min. Fig (e) and (f); $^*$P<0.05 and $^{***}$P<0.001 compared with values in animals treated with DMA alone.

involved the combination of the chloride HCO₃⁻ exchangers and the sodium hydrogen exchanger (NHE3), duodenum was perfused with an isotonic Cl⁻-free solution together with DMA. Perfusion with isotonic Cl⁻-free Na₂SO₄ plus DMA did not affect the mean luminal alkalinisation (Fig 8A). However, similar to the mannitol-perfusion experiments, a very good (P<0.001) linear correlation (y = 3.31–0.62x, $r^2$ = 0.79, n = 11) was found between the basal luminal alkalinisation and the Na₂SO₄ plus DMA-induced change in luminal alkalinisation (Fig 4E). The Na₂SO₄ plus DMA solution changed (P<0.05) net fluid flux from a value not different from zero to net fluid secretion (Fig 8B).

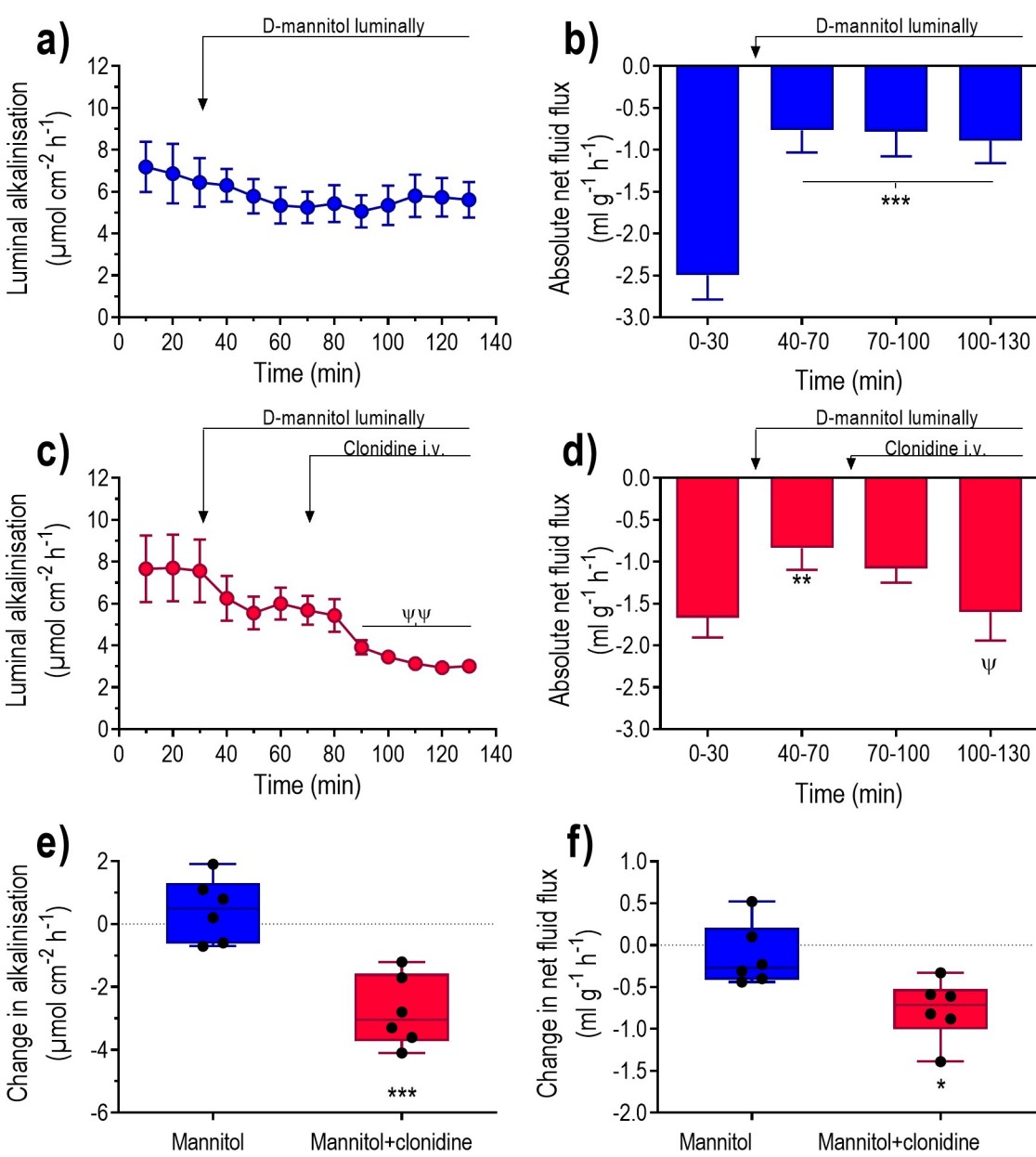

**Fig 7. The effects of clonidine on fluid flux and luminal alkalinisation in rat duodenum perfused with a NaCl-free solution.**
Duodenum was perfused with isotonic saline for 30 min and then with an isotonic D-mannitol solution for 100 min (Fig 1). Effects on luminal alkalinisation (a and c) and transepithelial net fluid flux (b and d) were determined in the absence and presence of i.v. infusion of clonidine at a dose of 10 μg kg⁻¹ h⁻¹. Net changes in (e) luminal alkalinisation and (f) transepithelial net fluid flux between 110–130 min and 50–70 min (D-mannitol plus clonidine vs. D-mannitol alone). Values are means ± SEM or box plots with all individual points. **P<0.01 and ***P<0.001 compared with basal values. ψP<0.05, ψψP<0.01 compared with values at 50–70 min. Fig (e) and (f); *P<0.05 and ***P<0.001 compared with values in animals treated with D-mannitol alone.

In animals perfused with the Cl⁻-free Na₂SO₄ plus DMA solution clonidine reduced luminal alkalinisation and abolished the Na₂SO₄ plus DMA-induced net fluid secretion (Fig 8C and 8D). The changes in luminal alkalinisation and the net fluid flux were both significantly greater in clonidine treated animals (Fig 8E and 8F).

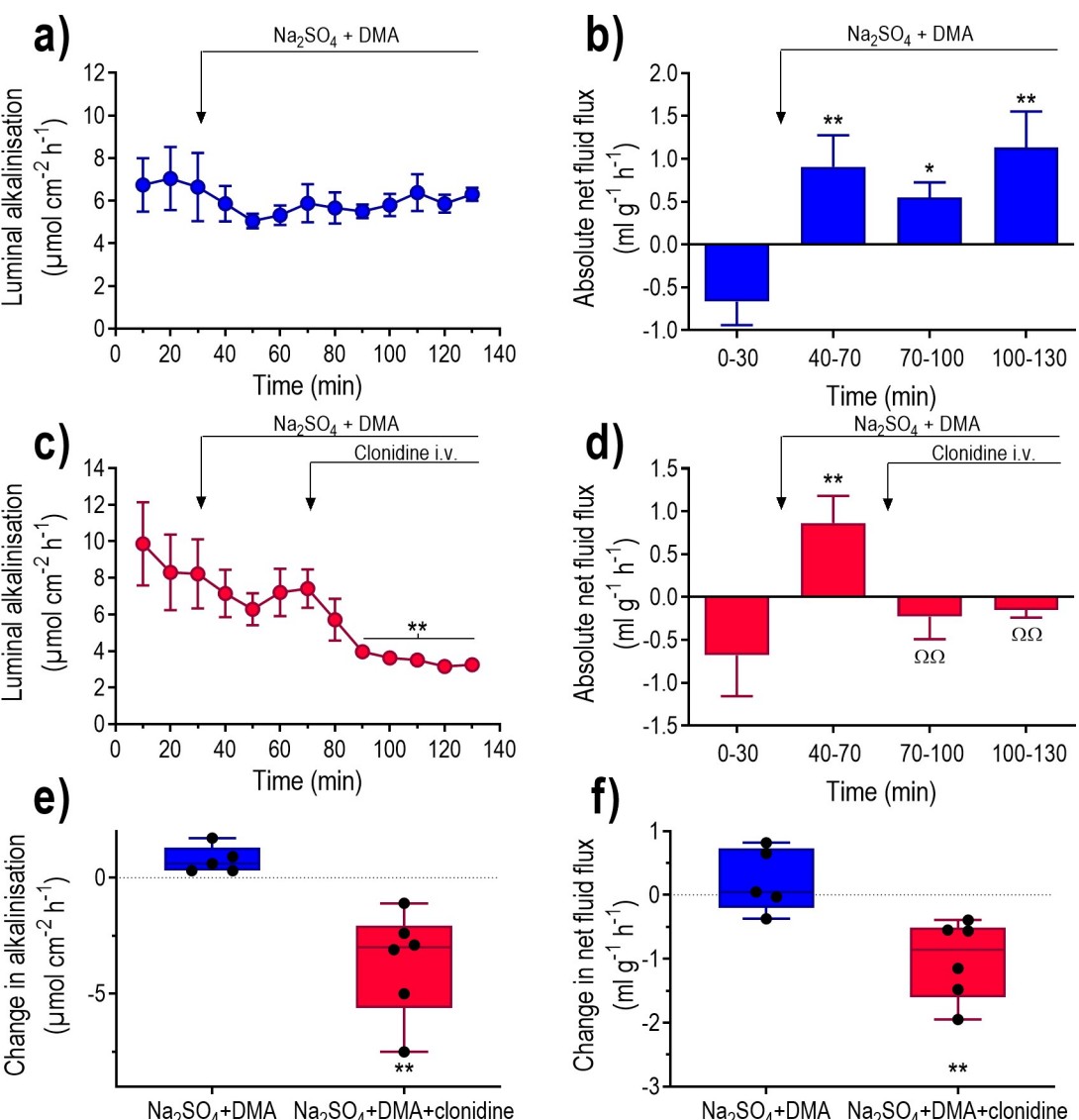

**Fig 8. The effects of clonidine on fluid flux and luminal alkalinisation in rat duodenum perfused with a Cl⁻-free + DMA solution.** Duodenum was perfused with isotonic saline for 30 min and then an isotonic $Na_2SO_4$ plus dimethylamiloride (DMA) solution for 100 min (Fig 1). Effects on (a and c) luminal alkalinisation and (b and d) transepithelial net fluid flux were determined in the absence and presence of i.v. infusion of clonidine at a dose of 10 µg $kg^{-1}$ $h^{-1}$. Changes in (e) luminal alkalinisation and (f) transepithelial net fluid flux between 110–130 min and 50–70 min ($Na_2SO_4$ plus DMA plus clonidine vs. $Na_2SO_4$ plus DMA alone). Values are means ± SEM or box plots with all individual points. Fig b) and d) *P<0.05 and **P<0.01 compared with values at 0–30 min. Fig (c) **P<0.01 compared with values at 40–70 min. Fig (c) ΦΦP<0.01 compared with compared with values at 50–70 min. Fig (e) and (f) **P<0.01 compared with animals treated with Na2SO4 plus DMA alone.

## Does α2-adrenoceptor stimulation affect the VIP-induced stimulation of electrolyte-fluid secretion?

There were reasons to believe that clonidine reduces luminal alkalinisation and induces net fluid absorption by inhibiting electrolyte fluid secretion. To investigate this possibility, we used vasoactive intestinal peptide (VIP), a well-recognised secretagogue, to stimulate electrolyte fluid secretion. In the first series of experiments we investigated the effects of clonidine in animals treated with VIP. VIP increased (P<0.001) luminal alkalinisation and induced

(P<0.001) net fluid secretion (Fig 9A and 9B). In VIP-treated animals clonidine decreased, but did not abolish, luminal alkalinisation (P<0.05) and reduced net fluid secretion (P<0.01).

In a second series of experiments we examined whether VIP could increase electrolyte fluid secretion in the presence of clonidine. Indeed, VIP increased (P<0.001) luminal alkalinisation and changed (P<0.001) net fluid flux from net fluid absorption to net secretion (Fig 9C and 9D). The net increase in luminal alkalinisation and the change in net fluid flux in response to VIP were virtually the same in animals treated with VIP alone as in those pre-treated with clonidine (Fig 9E and 9F). The net decrease in luminal alkalinisation and the change in net fluid flux in response to clonidine were virtually the same in animals treated with clonidine alone and in those pre-treated with VIP (Fig 9G and 9H).

### Does α2-adrenoceptor stimulation affect the nAChR-induced inhibition on fluid flux and luminal alkalinisation?

Clonidine may indirectly inhibit electrolyte fluid secretion by suppressing nicotinic acetylcholine receptors-activated (nAChR) secretomotor neurons that innervate the epithelium. If this were the case, treatment with hexamethonium would prevent clonidine from inhibiting luminal alkalinisation and inducing net fluid absorption. Administration of hexamethonium promptly and continuously reduced MABP (Fig 10A). Concomitantly, hexamethonium decreased luminal alkalinisation and augmented net fluid absorption (Fig 10B and 10C). A very good correlation was found between the basal luminal alkalinisation and the hexamethonium-induced decrease in luminal alkalinisation (y = 2.04–0.69x, $r^2$ = 0.77, P<0.001, n = 15), (Fig 10D). Intravenous infusion of clonidine to rats pre-treated with hexamethonium had no effect on MABP (Fig 10A), luminal alkalinisation, or net fluid flux (Fig 10E and 10F).

### Summary of treatment effects on fluid flux and luminal alkalinisation

Interventions and their effects on fluid absorption and luminal alkalinisation in the villi and crypt regions are summarized in Fig 11.

### Discussion

The aim of the present investigation was to shed further light on the mechanism by which α2-adrenoceptor stimulation inhibits luminal alkalinisation and induces net fluid absorption in the rat duodenum *in vivo*. More specifically, we wanted to answer if (i) the effects of clonidine on electrolyte and water transport were due to increased absorption or reduced secretion or a combination of both, (ii) if the effects of clonidine were sensitive to the removal of luminal $Cl^-$ and/or $Na^+$, and (iii) if clonidine exerted its effects via suppression of excitatory nicotinergic receptor-activated secretomotor neurons [3].

Clonidine at two doses induced the same inhibition of luminal alkalinisation and induction of net fluid absorption. The fact that the effects were markedly attenuated by the α2-adrenoceptor antagonist idazoxan, strongly suggests that clonidine exerts its effects via α2-adrenoceptors. An interesting observation was that clonidine was less effective in reducing luminal alkalinisation and augmenting net fluid absorption in rats with a low basal rate of alkalinisation or a high basal net fluid absorption, respectively, which most likely reflects a higher basal sympathetic tone to the duodenal segment in these rats.

Previous *in vivo* experiments in rodent duodenum have shown that ablation of $Cl^-/HCO_3^-$ exchangers (Slc26a6 or Slc26a3) reduces basal duodenal mucosal $HCO_3^-$ secretion, and that luminal perfusion with a $Cl^-$-free solution markedly reduces luminal alkalinisation [18, 30]. This suggests that $Cl^-/HCO_3^-$ exchangers play an important role in regulating duodenal mucosal $HCO_3^-$ secretion. The results in our study clearly showed that the magnitude of the $Cl^-$-free

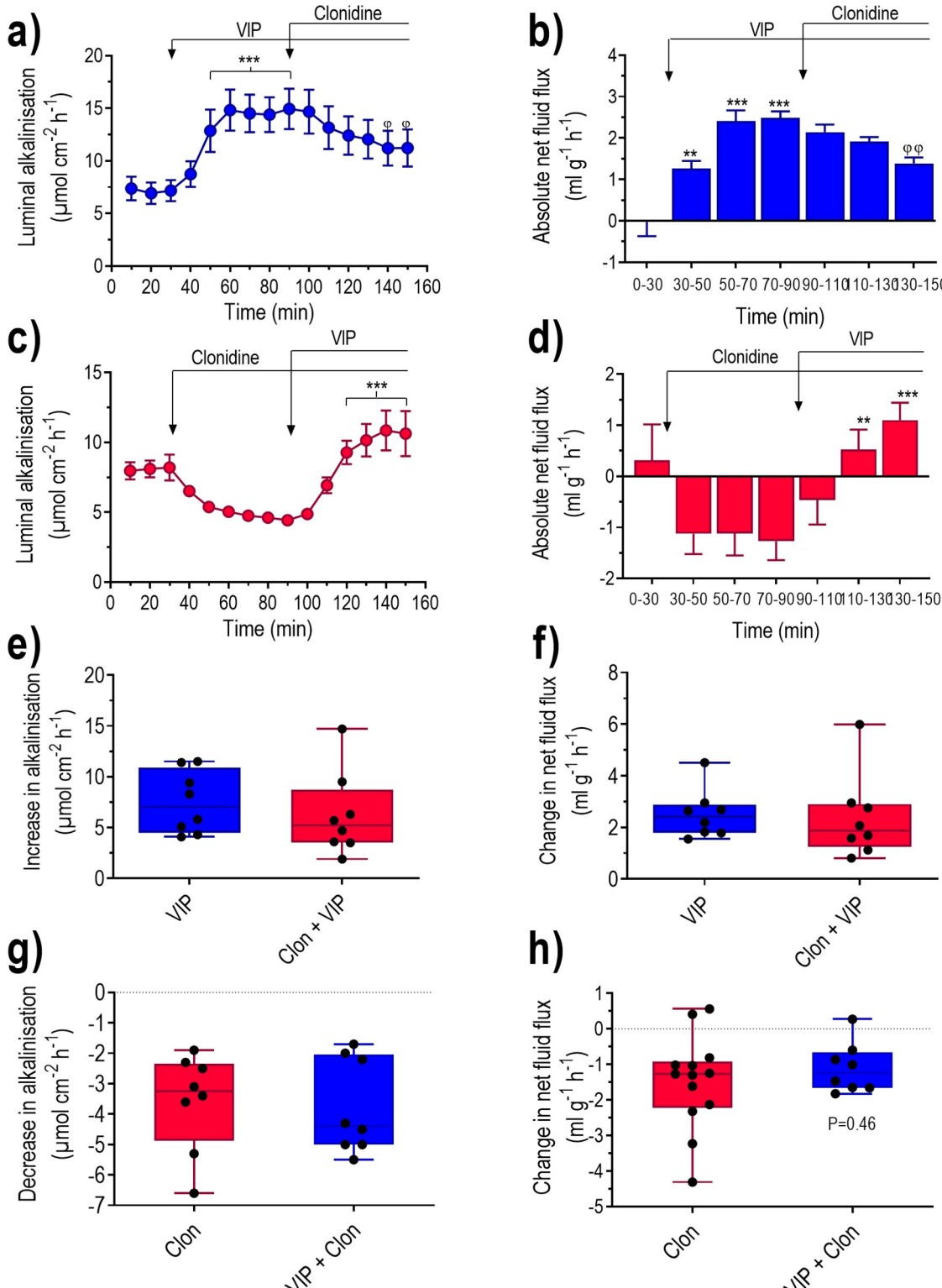

**Fig 9. The effects of clonidine administered before and after treatment with vasoactive intestinal peptide (VIP) on fluid flux and luminal alkalinisation in rat duodenum.** Duodenum was perfused with isotonic saline for 150 min. Effects on (a and c) luminal alkalinisation and (b and d) transepithelial net fluid flux determined with (a and b) VIP (i.v. 15 μg kg$^{-1}$ h$^{-1}$) from 30 min and clonidine (i.v. 10 μg kg$^{-1}$ h$^{-1}$) from 90 min, or with (c and d) clonidine from 30 min and VIP from 90 min. The (e) increase in luminal alkalinisation and the (f) change in net fluid flux in response to VIP alone (mean 70–90 min minus 0–30 min) and in

response to clonidine (mean 130–150 min minus 70–90 min). The (e) decrease in luminal alkalinisation and the (f) change in net fluid flux in response to clonidine alone (mean 70–90 min minus 0–30 min) and in response to VIP (mean 130–150 min minus 70–90 min). Values are means ± SEM or box plots with all individual points. Fig a-b. **P<0.01 and ***P<0.001 compared with basal values. ᵠP<0.05 and ᵠᵠP<0.01 compared with values at time points 70–90 min. Fig c-d. **P<0.01 and ***P<0.001 compared with values at time point 70–90. Fig e. *P<0.05 compared with VIP alone.

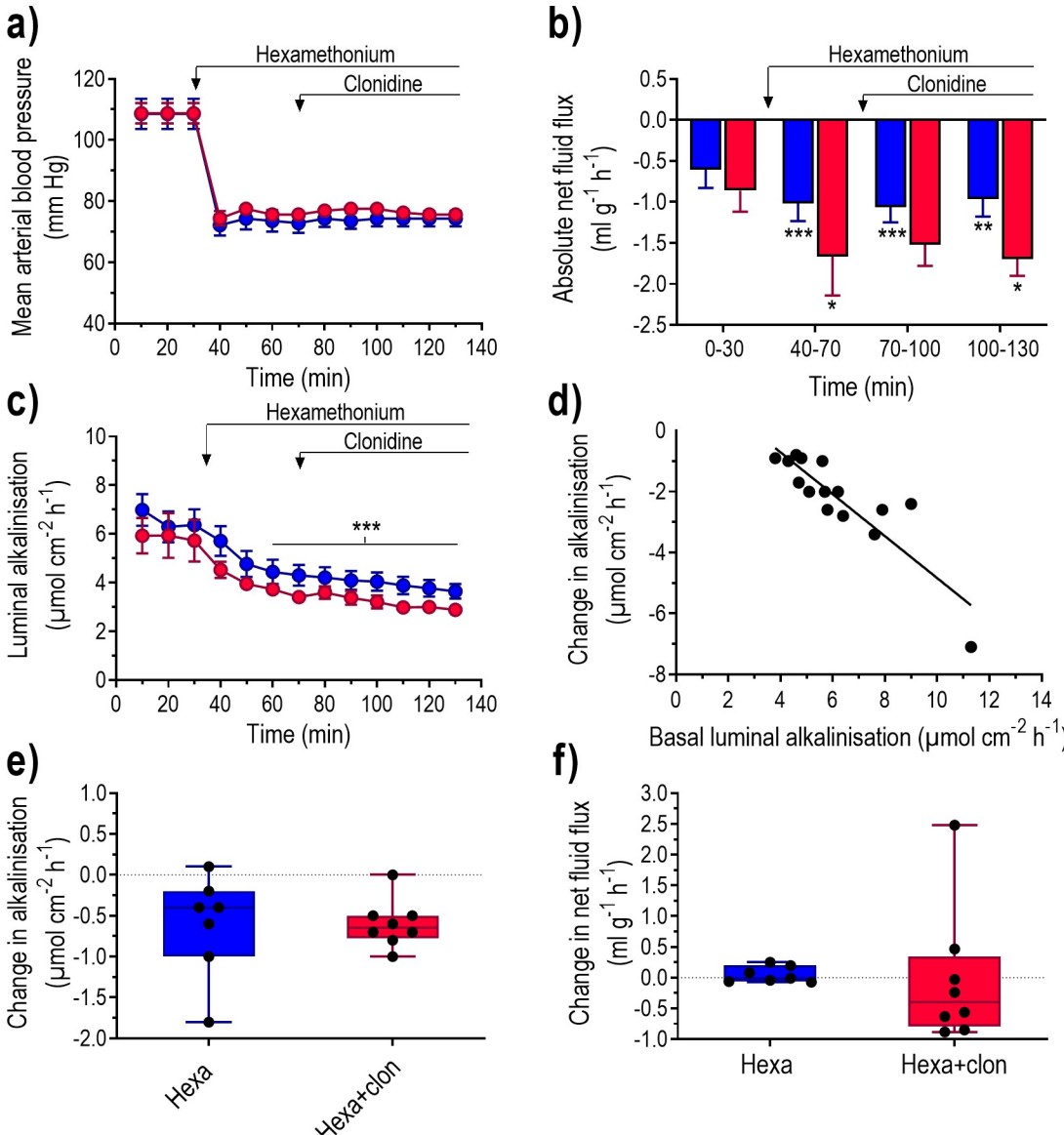

**Fig 10. The effect of clonidine administered after treatment with a non-selective nicotinic receptor inhibitor (hexamethonium) on blood pressure, net fluid flux and luminal alkalinisation in the rat duodenum.** Duodenum was perfused with isotonic saline for 130 min with hexamethonium (i.v. 10 mg kg⁻¹ h⁻¹) from 30 min followed by clonidine (i.v. 10 µg kg⁻¹ h⁻¹) from 70 min (Fig 1). Effects on (a) mean arterial blood pressure, (b) transepithelial net fluid flux, and (c) luminal alkalinisation with. The (d) relationship between the basal luminal alkalinisation and the changes in luminal alkalinisation in response to hexamethonium compared to baseline (0–30 min). Changes in (e) luminal alkalinisation and (f) transepithelial net fluid flux between 100–130 and 40–70 min in animals treated with hexamethonium alone and hexamethonium plus clonidine. Values are means ± SEM or box plots with all individual points. **P<0.01 and ***P<0.001 compared with basal values. ᵠP<0.05 and ᵠᵠP<0.01 compared with values at time point 70–90.

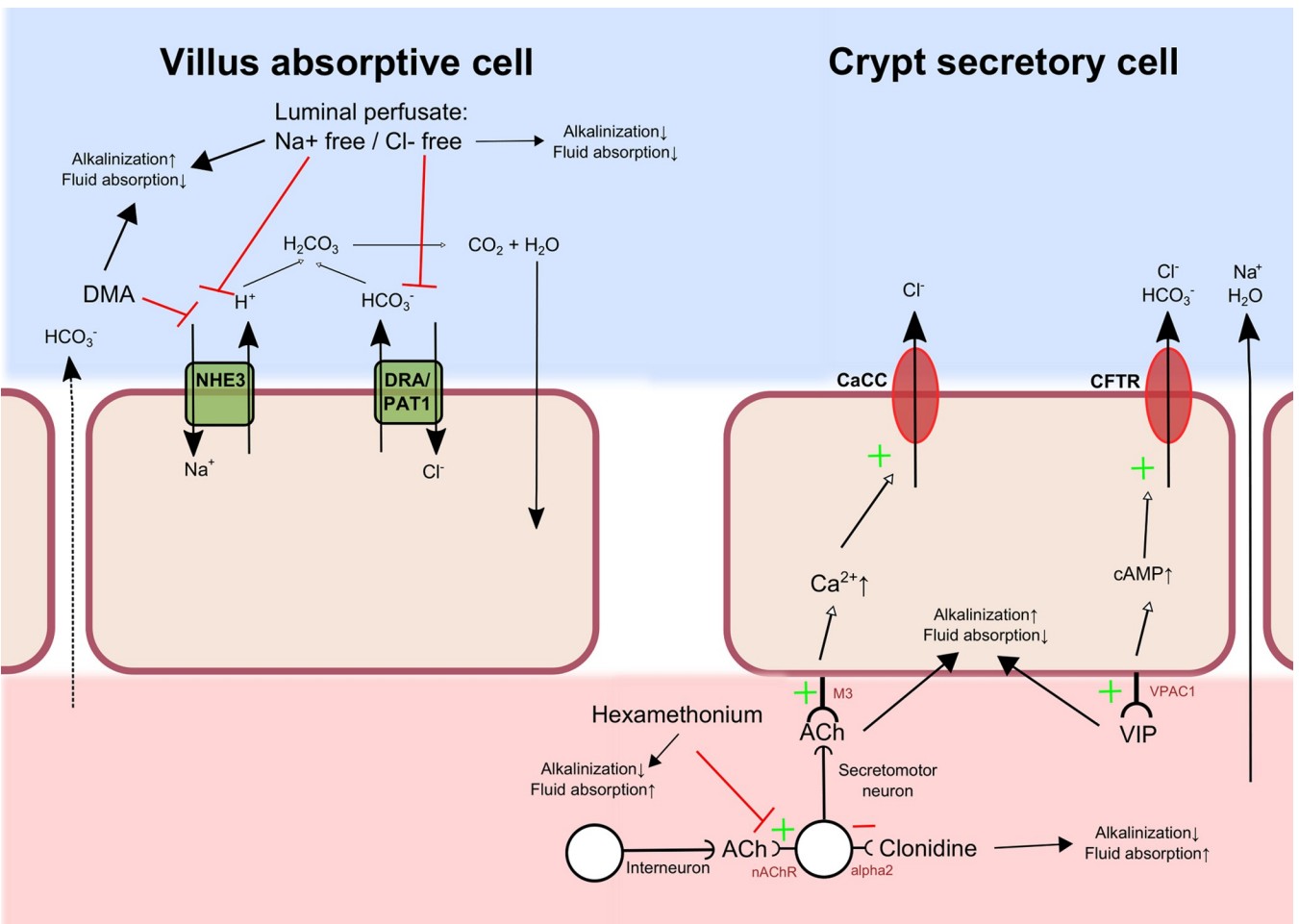

**Fig 11. A summary of the effects on duodenal fluid absorption and luminal alkalinisation in the villi and crypts.** Effects on rat duodenal fluid absorption and luminal alkalinisation of clonidine combined with sodium and/or chloride free luminal perfusates, luminal inhibition of the Na⁺/H⁺ exchanger (NHE3) with DMA, and intravenous administrations of the vasoactive intestinal peptide (VIP) or hexamethonium, a non-selective nicotinic receptor antagonist. DRA/PAT1—chloride anion exchanger, CaCC–calcium-activated chloride channels, CFTR—Cystic fibrosis transmembrane conductance regulator, Ach–acetylcholine, cAMP–Cyclic adenosine monophosphate, nAChR—Nicotinic acetylcholine receptor, alpha2—Alpha-2 adrenoceptor, VPAC1—Vasoactive intestinal polypeptide receptor 1, M3—Muscarinic M3 receptor.

solution-induced decrease in luminal alkalinisation correlated well with the basal rate of luminal alkalinisation. In other words, the greater the basal rate, the greater the decrease in alkalinisation. The great variation in basal duodenal luminal alkalinisation among animals could thus reflect different activity of the apical Cl⁻/ HCO₃⁻ exchangers. This in turn hint that the apical Cl⁻/ HCO₃⁻ exchangers are regulated and dependent on neurotransmitters, hormones, and paracrine factor. In fact, Tuo *et al.* (2006) [31] found that the PGE₂- and the carbachol-stimulated duodenal HCO₃⁻ secretion was reduced in Slc26A6-deficient mice *in vitro*. Other experiments *in vivo* in the rat duodenum show that the motility-induced stimulation of duodenal luminal alkalinisation is almost completely abolished by removal of luminal Cl⁻ [30].

The Cl⁻-free solution-induced decrease in luminal alkalinisation was associated with abolishment of net fluid absorption. This confirms previous findings in humans and rodents that luminal Cl⁻ and the presence of functional Cl⁻/HCO₃⁻ exchangers are required for a normal intestinal fluid absorption [32–34].

In the present study it is clearly shown that clonidine significantly reduced luminal alkalinisation and induced net fluid absorption also rats perfused with a Cl⁻-free solution, strongly suggesting a mechanism independent of luminal Cl⁻.

Luminal perfusion of the duodenal segment with the Na⁺-free solution, or inhibition of Na⁺/H⁺ exchange (DMA), increased luminal alkalinisation, confirming findings from rat and mouse [17, 18]. This probably occurs via inhibition of apical H⁺ efflux, which then unmasks the activity of Cl⁻/HCO₃⁻ exchangers and the CFTR channel. In contrast to the Cl⁻-free solution, no linear correlation was found between the basal rate of luminal alkalinisation and the Na⁺-free or DMA-induced increase in luminal alkalinisation. One explanation may be a predominance of Cl⁻/HCO₃⁻ exchangers over Na⁺/H⁺ exchangers in duodenum [35, 36]. Another explanation may be that the activity of apical Na⁺/H⁺ exchangers is less variable than that of the Cl⁻/HCO₃⁻ exchangers in our *in vivo* duodenal-perfusion model.

Luminal perfusion with the Na⁺-free (NMDG) or the DMA solution abolished net fluid absorption, in agreement with findings that the lack of luminal Na⁺, NHE3-gene knockout [34], or NHE3 inhibition by S1611 [18], reduce fluid absorption in the mouse small intestine. Clonidine may induce net fluid absorption and inhibit luminal alkalinisation by stimulation of basal NHE3, and/or by inhibiting the cAMP- and Ca²⁺-induced decreases in apical NHE3 activity [37]. Our data showed that clonidine effectively reduced luminal alkalinisation and induced net fluid absorption also in the absence of luminal Na⁺ and in the presence of DMA, suggesting a mechanism independent of Na⁺/H⁺ exchange.

The fact that the Cl⁻-free (Na₂SO₄) solution decreased luminal alkalinisation, while the Na⁺-free one (NMDG) increased it made us curious what would happen in response to luminal perfusion with the NaCl-free mannitol solution, or the Cl⁻-free plus DMA solution. It turned out that these two solutions had no significant effect on luminal alkalinisation. However, the change in luminal alkalinisation of the two treatments depended on the rate of basal luminal alkalinisation. In rats with a spontaneously low basal alkalinisation, the decrease in luminal alkalinisation induced by the removal of Cl⁻ was probably counterbalanced by the increase in alkalinisation brought about by the lack of luminal Na⁺, or by the inhibition of Na⁺/H⁺ exchange. Clonidine also decreased luminal alkalinisation in the absence of luminal NaCl, and luminal Cl⁻ plus inhibition of Na⁺/H⁺ exchange. The mannitol solution markedly attenuated the net fluid absorption while the Cl⁻-free plus DMA solution induced net fluid secretion. Clonidine augmented the net fluid absorption in mannitol-perfused animals and abolished the net fluid secretion in those perfused with Na₂SO₄ plus DMA. Taken together, these results further suggested that clonidine reduces luminal alkalinisation and induces net fluid absorption by a mechanism independent of Cl⁻/HCO₃⁻ and/ Na⁺/H⁺ exchange.

How then does clonidine reduce luminal alkalinisation and induce net fluid absorption? The most obvious explanation is that clonidine inhibits electrolyte fluid secretion. Most likely this occurs by inhibition of CFTR or other Cl⁻ channels in the crypt region of the epithelium, where α2-adrenoceptors are predominately expressed, both in rat jejunum and human duodenum [38, 39]. We tested this hypothesis by examining the ability of clonidine to inhibit the effects of VIP, which is normally found in nerve fibres in proximity to the duodenal epithelium [40, 41]. It has previously been shown that VIP stimulates duodenal mucosal HCO₃⁻ transport [42–44], as well as fluid secretion in animals with functional CFTR activity [27, 45]. In guinea pig jejunum *in vitro*, VIP appears to exert its stimulatory effect on secretion (short-circuit current), predominately via activation of VPAC1 receptors located in the mucosa and partly via an action on submucosal neurons [46].

If VIP and clonidine exert their actions on the same target cell, it seems reasonable to assume that clonidine would reduce the stimulatory effect of VIP, and that VIP would reduce the inhibitory action of clonidine. In the first series of experiments, clonidine reduced both the

VIP-induced increase in luminal alkalinisation and the increase in net fluid secretion, in agreement with data from rat jejunum *in vivo* [47]. However, in animals pre-treated with clonidine, the VIP-induced increase in luminal alkalinisation and net fluid secretion were not different from what was obtained with VIP alone. Furthermore, the net decrease in luminal alkalinisation and the change in net fluid flux in response to clonidine were virtually the same in animals treated with clonidine alone as in those pre-treated with VIP. It thus appears that clonidine does not affect the VIP-induced stimulation of secretion but rather the basal secretion, which may be regulated by a different neural mechanism, at least in rat duodenum.

The modest, if any, inhibitory effect of clonidine on the VIP-induced stimulation of electrolyte fluid secretion raised the possibility that clonidine inhibits electrolyte fluid secretion indirectly, possibly via activation of $\alpha_2$-adrenoceptors on cholinergic enteric secretomotor neurons. We know from previous experiments that the non-selective nicotinic receptor antagonist, hexamethonium, reduces basal luminal alkalinisation [25, 26], which was confirmed in the present study. The degree of inhibition of luminal alkalinisation by hexamethonium highly correlated to basal rates of luminal alkalinisation, which possibly reflects variation in secretomotor neuron activity to the epithelium. Furthermore, here we showed that hexamethonium augmented net fluid absorption by inhibiting secretion. If clonidine exerts its action solely by blocking the activity in these hexamethonium-sensitive nerves, it is reasonable to assume that it would have no effect in hexamethonium-treated rats. This turned out to be the case, which favours the notion that clonidine inhibits electrolyte-fluid secretion via suppression of the activity in excitatory secretomotor neurons, in line with findings in rat jejunum *in vivo* [48]. Furthermore, the data does not support a direct effect of clonidine on α2-adrenoceptors expressed on epithelial cells, as suggested from stripped ileal mucosa with the voltage clamp technique [49]. Most likely clonidine acts by reducing the release of acetylcholine by secretomotor neurons, which binds to muscarinic M3 receptors on the epithelial crypt cells (Fig 11). Acetylcholine causes intracellular $Ca^{2+}$ to increase (while VIP increases cAMP), thereby stimulating apical $Cl^-$ secretion via the calcium-dependent chloride channel, as well as basolateral $K^+$ secretion [50]. These effects on ion secretion (i.e., VIP and acetylcholine) act synergistically [51, 52], which may explain why clonidine was still active even at the high VIP doses used in this study. We currently have ongoing experiments including muscarinic receptor antagonist to verify this.

In conclusion, the potent α2-adrenoceptor agonist clonidine turned out to inhibit luminal alkalinisation and to induce fluid absorption or inhibit secretion, in the absence of either luminal $Cl^-$ or $Na^+$ or both, and in the presence of a $Na^+/H^+$ exchange inhibitor. Although clonidine slightly reduced the VIP-induced stimulation of luminal alkalinisation and net fluid secretion, it did not affect the magnitude of the VIP-induced fluid secretion. The suppressive effect of clonidine on luminal alkalinisation as well as its pro-absorptive action was abolished by nicotinic receptor blockade. Collectively, these *in vivo* results suggest that clonidine exerts its effects predominately via inhibition of fluid secretion due to suppression of excitatory nicotinergic receptor-activated secretomotor acetylcholine neurons and probably not by direct action on epithelial cells, at least not in the rat duodenum in vivo.

## Supporting information

**S1 Data.**
(XLSX)

## Author Contributions

**Conceptualization:** Olof Nylander.

**Data curation:** John Sedin.

**Project administration:** Olof Nylander, Markus Sjöblom, David Dahlgren.

**Validation:** David Dahlgren.

**Writing – original draft:** Olof Nylander.

**Writing – review & editing:** Markus Sjöblom, John Sedin, David Dahlgren.

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
