## [Decision Letter · Decision Letter 0]

31 Mar 2022

PONE-D-22-02793The effects of α2-adrenoceptor stimulation on luminal alkalinisation and net fluid flux in the rat duodenumPLOS ONE

Dear Dr. Dahlgren,

Thank you for submitting your manuscript to PLOS ONE. The manuscript has been evaluated by two experts in the field. After careful consideration, we feel that the manuscript has merit but does not fully meet PLOS ONE’s publication criteria as it currently stands. Therefore, we invite you to submit a revised version of the manuscript that addresses the concerns raised by the reviewers.

We look forward to receiving your revised manuscript.

Kind regards,

Alexander G Obukhov, Ph.D.

Academic Editor

PLOS ONE

Journal Requirements:

2. To comply with PLOS ONE submissions requirements, in your Methods section, please provide additional information regarding the experiments involving animals and ensure you have included details on (1) methods of sacrifice, (2) methods of anesthesia and/or analgesia, and (3) efforts to alleviate suffering

Reviewers' comments:

Reviewer's Responses to Questions

**Comments to the Author**

1. Is the manuscript technically sound, and do the data support the conclusions?

Reviewer #1: Yes

Reviewer #2: Partly

2. Has the statistical analysis been performed appropriately and rigorously? 

Reviewer #1: Yes

Reviewer #2: No

3. Have the authors made all data underlying the findings in their manuscript fully available?

Reviewer #1: Yes

Reviewer #2: No

4. Is the manuscript presented in an intelligible fashion and written in standard English?

Reviewer #1: Yes

Reviewer #2: Yes

5. Review Comments to the Author

Reviewer #1: Reviewer Comments:

The article entitled “The effects of α2-adrenoceptor stimulation on luminal alkalinization and net fluid flux in the rat duodenum” sought to elucidate the mechanism by which α2-adrenergic receptor stimulation attenuates luminal alkalinization and augments net fluid absorption in the duodenum. The in vivo approach described herein systematically evaluated luminal alkalinization and net fluid absorption in the rat duodenum by perfusing different solutions in the presence and absence of α2-adrenergic receptor agonist. While it is obvious that a tremendous amount of time, effort, and resources were put into this work, some major concerns prevent me from recommending this article for publication.

Major Concerns

1. There is no stated relevance or significance mentioned in the Abstract, Introduction or Discussion. It would be nice to know why these receptors are important, their role in disease onset/progression, or other notable involvement.

2. Additionally, the article is mostly observational and supports the reports of others. How has this work moved the field forward? What needs to be done next? These questions will need to be discussed in the introduction and expanded on in the Discussion.

3. It would be helpful to have a figure that summarizes each perfusion condition over time. It could be put in the methods section.

4. It would also be useful to have a final figure depicting the conclusions and displaying the events inside and outside the cell.

Minor Concerns

1. The article needs further proofreading and editing by a native English-speaking individual.

Reviewer #2: The paper by Sjöblom and collaborators aims to understand the role of alpha2-adrenoceptor agonism on fluid and acid-base transport in the duodenum. They employ a classical and elegant approach to achieve these goals, using in vivo measurements of fluid flow and luminal alkalinization/acidification. However, a number of concerns in my view must be addressed before the paper is rendered publishable.

Major points:

1. presentation of the results - Up to the very last two subsections of the Results section, the text is very difficult to follow. It basically consists of an expanded list of expanded protocols (e.g. "clonidine +/- idaxozan", "Cl- free perfusion" etc) whose inner logic to address the issues the study aims to understand is not evident. The authors should thoroughly rewrite these subsections following the pattern they themselves used in the las two subsections, by briefly explaining at the beginning why they did those experiments and, at their ends, briefly explain which mechanistic conclusions can be drawn from the results. Also, the heading of each figure legend should be thoroughly expanded, as they should be self explanatory.

2. baseline levels - in many figures already the early baseline recordings are different between reference and text groups. The origin of such differences is not clear and are a big concern to the appropriate interpretation of the data (see point 3, below), as from what I could gather from the methods, one should not expect these differences, as the infusion of the drugs occurs acutely during the recordings, i.e. there is no pretreatment of the animals prior to the surgery that could generate these differences in baseline. Though not addressing this directly, the authors seem to have been concerned by that, considering they present various correlation graphs with data from the individual experiments. However, I did not see how these correlation graphs are helpful to their claims. As they are dealt with at present in the manuscript, I believe they could be withdrawn. If the authors choose to maintain them, then I believe they should be presented in the figures that present the summary data as they appear, not lumped together in a single figure. The fact that the authors state that the animals in the study weighed from 190 to 503 g is also a source of concern. Could this be a source of such big differences in baseline values, as such dispersion in weights suggest a big difference in ages as well?

3. Statistics - I think the use of two-way ANOVA as the main statistical approach to analyze the data would deeply strengthen the authors' claims, as this would allow the authors to answer questions such as: "does time affect the size of the effect?"; "does [e.g. clonidine] treatment affect the variable?"; and "does the treatment affect (e.g. anticipate) the effect in time?" [i.e. interaction between treatment and time factors], already from the summary main results of the two-way ANOVA.

4. Systemic infusion of drugs - Though I understand that due to the nature of the model a targeted manipulation of adrenergic terminals specifically in the enteric nervous system is very difficult, one cannot escape the fact that various other systems and circuits could have been affected by the drugs the authors infused in the animals. For instance, hexamethonium is a general ganglionic blocker, and thus ganglionic parasympathetic synapses certainly were also affected by that. Considering that the parasympathetic inputs are generally considered to be the most intense autonomic modulators of enteric activity, the authors must consider this limitation of their study in the discussion section as well as discuss how could a general parasympathetic effect be ruled out in the hexamethonium experiments.

Minor points:

1. How many animals were used in total and what was their age range?

2. The authors present the luminal alkalinization data in units of umol . cm-2 . h-1. However, the reference they cite for the method presents the data as umol . cm-1 . h-1. Which is the correct unit? If the authors indeed did measure alkalinization flux per square area, how did they account for the increase in area due to luminal folds ans vilosities in the duodenum?

3. As absolute fluid flux data are presented as a function of time, it would make more sense if the data were presented as a cartesian plot with time as the x-axis, as the authors do for alkalinization flux, rather than presenting them as bar charts.

6. PLOS authors have the option to publish the peer review history of their article (what does this mean?). If published, this will include your full peer review and any attached files.

Reviewer #1: No

Reviewer #2: No

---

## [Author Response · Author response to Decision Letter 0]

25 May 2022

We would like to thank the reviewers for their careful evaluation of our manuscript and for providing constructive criticisms and suggestions. We have tried to address all the points raised by the reviewers. We think that this has improved the overall quality of the manuscript and we hope that it will be considered for publication in PLOSone. 

We hereby provide a detailed response to each of the questions raised by the reviewers.

Reviewer #1: Reviewer Comments:

The article entitled “The effects of α2-adrenoceptor stimulation on luminal alkalinization and net fluid flux in the rat duodenum” sought to elucidate the mechanism by which α2-adrenergic receptor stimulation attenuates luminal alkalinization and augments net fluid absorption in the duodenum. The in vivo approach described herein systematically evaluated luminal alkalinization and net fluid absorption in the rat duodenum by perfusing different solutions in the presence and absence of α2-adrenergic receptor agonist. While it is obvious that a tremendous amount of time, effort, and resources were put into this work, some major concerns prevent me from recommending this article for publication.

Major Concerns

1. There is no stated relevance or significance mentioned in the Abstract, Introduction or Discussion. It would be nice to know why these receptors are important, their role in disease onset/progression, or other notable involvement.

Response: On your suggestion we have updated the abstract, introduction and discussion to make the relevance of our work more evident. We want to emphasise that this is a basic physiology investigation of sympathetic regulation of intestinal functions based on data from the rat in vivo model.

2. Additionally, the article is mostly observational and supports the reports of others. How has this work moved the field forward? What needs to be done next? These questions will need to be discussed in the introduction and expanded on in the Discussion.

Response: The end-effects of adrenergic stimulation on the intestines is well known for a long time, where it acts as an anti-secretory agent. However, the exact mechanisms by which it alters secretory activity in the epithelium is not well established. Rather, are the effects on fluid flux mediated by an increased water absorption and/or by a reduced water secretion? The same goes for luminal alkalinisation, which we know is reduced by adrenergic stimulation. However, we do not know for sure if this is regulated primarily by effects on H+ and/or HCO3- secretion, what transporter that is primarily involved, or what neural pathways. Our aim was to shed some light on these questions by employing a new stratery, namely, an in vivo model duodenal rat model. Combined, we have strong data suggesting that the effect of α2-adrenoceptor stimulation is on the level of secretion. In the updated manuscript we have included work on the same issue from other groups, highlighted the novelty in our work, as well as included future recommendations of studies. We hope that these changes is to your satisfaction. 

3. It would be helpful to have a figure that summarizes each perfusion condition over time. It could be put in the methods section.

Response: Thanks for this great suggestion. Fig 1 describing the experimental method and treatment groups has been added to the manuscript.

4. It would also be useful to have a final figure depicting the conclusions and displaying the events inside and outside the cell.

Response: Also a great suggestion. On your recommendation we have added a schematic illustration (Fig 11) showing the different IV and luminal treatments and their corresponding effects on fluid absorption and luminal alkalinisation in the villus and crypt cells.

Minor Concerns

1. The article needs further proofreading and editing by a native English-speaking individual.

Response: On your suggestion, our manuscript has now been language edited by a professionally trained expert, and it has carefully been proofread

Reviewer #2: The paper by Sjöblom and collaborators aims to understand the role of alpha2-adrenoceptor agonism on fluid and acid-base transport in the duodenum. They employ a classical and elegant approach to achieve these goals, using in vivo measurements of fluid flow and luminal alkalinisation/acidification. However, a number of concerns in my view must be addressed before the paper is rendered publishable.

Major points:

1. Presentation of the results - Up to the very last two subsections of the Results section, the text is very difficult to follow. It basically consists of an expanded list of expanded protocols (e.g. "clonidine +/- idaxozan", "Cl- free perfusion" etc) whose inner logic to address the issues the study aims to understand is not evident. The authors should thoroughly rewrite these subsections following the pattern they themselves used in the las two subsections, by briefly explaining at the beginning why they did those experiments and, at their ends, briefly explain which mechanistic conclusions can be drawn from the results. Also, the heading of each figure legend should be thoroughly expanded, as they should be self explanatory.

Response: We fully agree with your suggestion. We have updated the heading in the results section to better reflect the content. We have also added an explanatory text in the beginning of each results section to introduce the reader to the rationale behind the experiments and what we wanted to investigate. In addition, we have updated all figure texts to be more detailed and self-explanatory.

2. baseline levels - in many figures already the early baseline recordings are different between reference and text groups. The origin of such differences is not clear and are a big concern to the appropriate interpretation of the data (see point 3, below), as from what I could gather from the methods, one should not expect these differences, as the infusion of the drugs occurs acutely during the recordings, i.e. there is no pretreatment of the animals prior to the surgery that could generate these differences in baseline. Though not addressing this directly, the authors seem to have been concerned by that, considering they present various correlation graphs with data from the individual experiments. 

Response: We do not agree. According to us, and also based on our long experience working with in vivo models, the variability in baseline physiological parameters (e.g. permeability, blood pressure, bicarbonate secretion, ion and fluid flux) is something normal and expected, and in our opinion far from a “big concern” regarding interpretation of the data. On the contrary, this is one major finding of this study, where we are (for the first time as far as we can tell) able to show that the degree of effect of different luminal and intravenous treatments are directly related to the baseline bicarbonate secretion value of that animal; A high basal bicarbonate secretion value results in a big drop with treatment. This is thouroughly discussed in the manuscript already.

However, I did not see how these correlation graphs are helpful to their claims. As they are dealt with at present in the manuscript, I believe they could be withdrawn. If the authors choose to maintain them, then I believe they should be presented in the figures that present the summary data as they appear, not lumped together in a single figure. 

Response: We believe that that correlations should be placed in one figure. This is because this enables the interpretation of how basal bicarbonate secretion relates to a reduction in bicarbonate secretion for different types of treatment. By separating them, this analysis is made more complicated. By combinaing the graphs in one figure it is evident that the differences in basal luminal alkalinisation is related to the basal activity of the apical Cl/HCO3 exchanger, and not to the apical Na/H exchanger, as only removal of luminal Cl induced a linear reduction in luminal alkalinization, while the removal of Na+ resulted in random changes. 

The fact that the authors state that the animals in the study weighed from 190 to 503 g is also a source of concern. Could this be a source of such big differences in baseline values, as such dispersion in weights suggest a big difference in ages as well?

Response: When reviewing our data we noticed that this was a typo. The weight interval was 260-389 (316±27, n=105). Nontheless, our long experience working with the duodenal rat SPIP model have not indicated that age and weight, within a reasonable range, affects bicarbonate secretion, fluid flux and permeability.

3. Statistics - I think the use of two-way ANOVA as the main statistical approach to analyze the data would deeply strengthen the authors' claims, as this would allow the authors to answer questions such as: "does time affect the size of the effect?"; "does [e.g. clonidine] treatment affect the variable?"; and "does the treatment affect (e.g. anticipate) the effect in time?" [i.e. interaction between treatment and time factors], already from the summary main results of the two-way ANOVA.

Response: We appreciate the suggestion to use the two-way ANOVA. However, guidance from our statistician recommend us to validate the differences as is presented in this manuscript. 

The rational of this is; In most of comparations we evaluate changes within one group. We start each experiment collecting basal values at steady state, and then compare these to new steady state values after different iv and luminal treatments. To test the difference within one group, a one-factor repeated measure ANOVA was used followed by Tukey's Multiple Comparison Test. Differences between two groups is tested in students t-test, and when multiple comparisons are needed an unpaired ANOVA is used (i.e. a two-factor repeated measures ANOVA). 

Hope that this better describes how the statistical analysis was performed and that you may support our approach.

4. Systemic infusion of drugs - Though I understand that due to the nature of the model a targeted manipulation of adrenergic terminals specifically in the enteric nervous system is very difficult, one cannot escape the fact that various other systems and circuits could have been affected by the drugs the authors infused in the animals. For instance, hexamethonium is a general ganglionic blocker, and thus ganglionic parasympathetic synapses certainly were also affected by that. Considering that the parasympathetic inputs are generally considered to be the most intense autonomic modulators of enteric activity, the authors must consider this limitation of their study in the discussion section as well as discuss how could a general parasympathetic effect be ruled out in the hexamethonium experiments.

Response: Hexamethonium reduced blood pressure by suppressing sympathetic activity in autonomic ganglions, which is corroborated by that clonidine had no additional effect on blood pressure reduction. The effect of hexamethonium on alkaline secretion could therefore be because of an inhibition of parasympathetic cholinergic activity in the gut, although we cannot exclude the influence of other transmitters than acetylcholine. Furthermore, we cannot see any other way that the parasympathetic nervous system may directly, or indirectly, affect enteric functions.

Minor points:

1. How many animals were used in total and what was their age range?

Response: In total 105 rats was used in this study. We do not order rats by age but by weight. By reviewing the weight age curve from the breeder the age range is estimated to 7 to 12 weeks.

2. The authors present the luminal alkalinization data in units of umol . cm-2 . h-1. However, the reference they cite for the method presents the data as umol . cm-1 . h-1. Which is the correct unit? If the authors indeed did measure alkalinization flux per square area, how did they account for the increase in area due to luminal folds ans vilosities in the duodenum?

Response: Luminal alkalinisation was reported as umol/cm2/h. Indeed, by accounting for fold and villi, a lower flux value would result. However, we were interested in change in luminal alkalinisation in response to different treatments, rather than to investigate absolute values. As such, any unit could have been selected (e.g. umnol/g/h), and it would not change the interpretation of our results. The reason for us presenting data as per cm2 is because we have previously experienced that different strains of rat have different luminal diameter of the duodenum. A value describing secretion per area is thus more appropriate in our opinion.

3. As absolute fluid flux data are presented as a function of time, it would make more sense if the data were presented as a cartesian plot with time as the x-axis, as the authors do for alkalinization flux, rather than presenting them as bar charts.

Response: We agree that presenting data in the same way as for alkalinisation would have been ideal. However, due to the very small amount of water secreted (weight change) during each 10 min interval, an individual measurement reflects randomness in sampling time (in the order of seconds, or one drop to the next leaving the segment) rather than an actual fluid flux. To partly avoid this methodological issue we present data in bar charts where fluid flux from 3 measurements are combined.

---

## [Decision Letter · Decision Letter 1]

19 Jul 2022

PONE-D-22-02793R1Effects of α2-adrenoceptor stimulation on luminal alkalinisation and net fluid flux in rat duodenumPLOS ONE

Dear Dr. Dahlgren,

Thank you for submitting your revised manuscript to PLOS ONE. Before the manuscript can be accepted please add the error bars in Figure 2F and superimpose all bar graphs with the dot plots to show the distribution of raw data. 

We look forward to receiving your revised manuscript.

Kind regards,

Alexander G Obukhov, Ph.D.

Academic Editor

PLOS ONE

Journal Requirements:

Reviewers' comments:

Reviewer's Responses to Questions

**Comments to the Author**

1. If the authors have adequately addressed your comments raised in a previous round of review and you feel that this manuscript is now acceptable for publication, you may indicate that here to bypass the “Comments to the Author” section, enter your conflict of interest statement in the “Confidential to Editor” section, and submit your "Accept" recommendation.

Reviewer #2: All comments have been addressed

2. Is the manuscript technically sound, and do the data support the conclusions?

Reviewer #2: Yes

3. Has the statistical analysis been performed appropriately and rigorously? 

Reviewer #2: Yes

4. Have the authors made all data underlying the findings in their manuscript fully available?

Reviewer #2: Yes

5. Is the manuscript presented in an intelligible fashion and written in standard English?

Reviewer #2: Yes

6. Review Comments to the Author

Reviewer #2: The manuscript has been thoroughly revised and it is much improved in its readability. I congratulate the authors on their hard work and also on their thoughtful responses to this reviewer's comments on the original manuscript. The new summary figure in the discussion section is also very helpful.

7. PLOS authors have the option to publish the peer review history of their article (what does this mean?). If published, this will include your full peer review and any attached files.

Reviewer #2: No

---

## [Editor Report · Decision Letter 2]

4 Aug 2022

Effects of α2-adrenoceptor stimulation on luminal alkalinisation and net fluid flux in rat duodenum

PONE-D-22-02793R2

Dear Dr. Dahlgren,

We’re pleased to inform you that your manuscript has been judged scientifically suitable for publication and will be formally accepted for publication once it meets all outstanding technical requirements.

Kind regards,

Alexander G Obukhov, Ph.D.

Academic Editor

PLOS ONE

---

## [Editor Report · Acceptance letter]

16 Aug 2022

PONE-D-22-02793R2 

Effects of a2-adrenoceptor stimulation on luminal alkalinisation and net fluid flux in rat duodenum 

Dear Dr. Dahlgren:

I'm pleased to inform you that your manuscript has been deemed suitable for publication in PLOS ONE. Congratulations! Your manuscript is now with our production department. 

Kind regards, 

on behalf of

Dr. Alexander G Obukhov 

Academic Editor

PLOS ONE